# Economic and Spatial Restructuring in the Aras Economic Zone: The Impact of Cross-Border Cooperation

Hamid Jafarzadeh and Yangdong Feng *

Department of Urban and Rural Planning, School of Architecture and Fine Arts, Dalian University of Technology, Dalian 116024, China
* Correspondence: yangdongfeng@dlut.edu.cn

**Abstract:** Cross-border cooperation is critical for regional development capabilities. China and Iran are increasingly strengthening their partnership with the Belt and Road Initiative. This study constructs a regional socio-spatial development index system from the perspective of locals and establishes a DPSIR model with five layers: drivers, pressures, states, impacts, and responses. Bottom-up regional analyses (economic, social, and spatial) were performed to explore local perspectives on cross-border cooperation and assess the possible value system of development in a specific region. Based on 41 quantitative indicators and a genetic algorithm, causal links between economic changes and spatial restructuring were determined and verified. The findings show that cross-border cooperative growth is possible in the research region. Due to regional heterogeneity, excessive pressure on states, and evolving imbalances, we found imbalanced development inside and between sections. Additionally, foreign direct investment enhances cross-border cooperation, which can preserve and develop local economies. Additionally, this study provides suggestions and references for cross-border cooperation opportunities, challenges, and decision-making.

**Keywords:** cross-border cooperation; BRI-driven FDI; strategic planning; local approach; regional planning; economic change

## 1. Introduction

Planning for a region's growth potential and international collaboration necessitates a comprehensive assessment of regional development. Cross-border cooperation (CBC) has become an increasingly important aspect of international cooperation in recent years, meaning that it is imperative that regional development prospects are examined from this perspective.

Understanding regional development and spatial interaction difficulties, as well as the balance between strategic initiatives and more localized dynamics and how these factors are impacted by international strategies, is essential for conducting a project-based grounded evaluation [1,2]. Compared to the vast amounts of ecological, infrastructure, and numerical data used by planners, spatially categorized, social, and perceptual data are insufficient [3].

To acquire socio-perceptual information on housing and ecological practices, data from elsewhere in the literature [3–5] have been used. The use of local-perceptual data not only improves our understanding of how spatial restructuring features contribute to local well-being [6] but also sheds light on how social factors like local diversity and interactions influence the strategic provision Belt and Road Initiative (BRI)-driven Foreign Direct Investment (FDI). Sustainable economic growth depends on capital flows [7].

The majority of the existing literature analyzes spatial restructuring and economic changes through sectorial patterns and regional City-to-City (C2C) cooperation. Studies have evaluated optimal governmental investments to balance supply, demand, and adjustment mechanisms during challenging periods [8,9], e.g., the COVID-19 pandemic, and understand governmental responses to local issues [10]. The COVID-19 pandemic yielded

concerns about regional concentration and a lack of robustness and adaptability in supply networks [11].

Researchers have focused on intergovernmental dynamics while also basing their assessments on a series of interviews with senior officers from multinational corporations as well as urban planners and government officials. CBC's strategic planning, FDI, and local perception have all contributed to spatial restructuring dynamics that have not been adequately examined in the context of a Special Economic Zone (SEZ), ignoring local interests, inner-SEZ assessments, and distinctive characteristics. International cooperation usually brings about conflicts of interest because project-level studies are neglected over policy and strategic planning. CBC is challenging and unproductive if local capabilities and needs are not taken into consideration in top-down plans. Long-term international cooperation requires the consideration of how economic changes and spatial restructuring processes occur as a result of CBC. It also requires the consideration of how local conflicting interests influence regional development strategies.

The purpose of this study was to examine regional spatial performance disparities and BRI-driven economic changes since 2013 to determine whether regional imbalances increase or decrease as economic change occurs, as well as whether local responses to spatial restructuring favor or oppose planned B&R initiatives in the Aras Free Trade-Industrial Zones (AFTZ) in Iran.

The remainder of this paper is structured as follows: Section 2 examines the existing literature and outlines a theoretical framework to explain spatial disparities caused by FDI. Section 3 contains a discussion on local responses to CBC. Section 4 describes the study area, methodology, and data sources we used. Results and empirical findings are outlined in Section 5. The main findings of the study are summarized in Section 6, and conclusions are drawn in Section 7.

## 2. Literature Review

Understanding economic activity and social environments is crucial for sustainable development [12,13]. Community development involves people identifying and expressing their needs, addressing common problems, and influencing decisions that affect their lives [14]. Developed areas face complex, ongoing, and continuously changing challenges [15]; rapid and irregular growth [16]; exacerbated environmental and social issues [17], such as housing and immigration [18]; and population shrinkage [19]. The location of settlements was previously determined by their availability of water reservoirs, social and religious aspects [15], physical aspects, capital city, geographical features [20], and the climate [17], not necessarily global trade and economy [21].

SEZs contribute significantly to economic growth [6]. Studies on SEZs [22] have assessed relations between FDI on spatial location and sectoral patterns over time, as well as inner funds for localized initiatives and outside of SEZs regions [23] and interactions between local and central authorities, to better understand regional cooperation politics [2], i.e., C2C cooperation. However, the effects of restructuring [24] indicate that an over-reliance on financial incentives may make it difficult for SEZs to connect with the regional economic context [25,26]. The prediction of spatial growth is challenging since it involves factors such as economic growth, population growth, consumption, etc., which are hard to predict over long periods [27], particularly since global challenges (like the COVID-19 pandemic) complicate the optimization of local and national cooperation [9,10].

Consequently, governance is becoming more crucial as cities are taking on more financial, social, and political responsibilities. Economic growth and diversification [2] into connected activities lead to regional integration [28], resulting in stronger economic links between regions [29]. Spatial planning, as an important part of state and society relations, is linked to decision-making and topics including economic development [24], climate change, urban green structure design, housing features, and young people's social mobility [3,4,27,30]. In some countries, spatial planning has long been used as a major tool of government intervention in urban development [31].

Spatial connections can depict not only the complexity of a region's socioeconomic links but also the overall arrangement of the region [30]. Heterogeneity, regional disparities [32], and variations in land use are some of the primary drivers for conflict [33,34]. At several levels, various administrative entities need to adopt a cohesive strategy for spatial planning and environmental management [13]. However, such integration is expected to result in a more simple planning approach that considers both potential consequences and local expansion [35]. To enhance regional economic coordination, the advanced industrial structure is an integral component of economic ties [29].

Economic developments and their spatial manifestation on urban spatial structure are crucial to economic growth [36,37], especially with an increase in international events for competitiveness in the globalized economy [24,38–40]. Regional authorities are increasingly arguing that giant events benefit the local community in order to justify costly mega-projects [39]. Regional economic policy is significantly impacted by regional disparities as well as their short- and long-term changes [32].

With the increase in competitiveness for international events in the globalized economy [38–40] and the increase in CBC, SEZs are important spatial carriers for BRI construction and drivers of inclusive cooperation [6]. The growth of SEZs requires proactive ties with local economies as well as increased local government control over important state institutions without compromising the legitimacy or mission of the state [22]. However, as the effects of restructuring [24] indicate, the reliance on financial incentives may make it difficult for SEZs to connect with regional economies [41].

Previous studies have stressed the importance of urban spatial structure [37] for economic growth in an intra-metropolitan context. Policies and strategic planning are typically prioritized over project-level studies, resulting in conflicts between various international and regional developmental policymakers [42]. Top-down strategies will deem bottom-up initiatives ineffective if they do not consider local capabilities and needs [32].

For this paper, we made several key contributions to spatial restructuring theory. First, an innovative method for measuring economic changes and spatial restructuring to assess environmental impacts and CBC was developed. Second, as part of the BRI strategies' sustainable implementation, a creative spatial planning and territorial governance process based on local responses for strategic and common regional planning was established. This paper presents an effective and comprehensive framework for identifying CBC factors. Additionally, in rationally assuming that learning alogithims will be used in future research, our study improved upon the previous limitations of the model [8]. The selection of the appropriate indicators still constitutes a research gap in the literature.

A comprehensive analysis of the development potential of CBC in SEZs is provided in this study, which enriches the research methods for evaluating its developmental potential. In addition to providing scientific and technological support, it also supports the economic development of the Aras SEZs and existing industrial bases within the BRI context.

## 3. Theoretical Framework

According to the current theoretical approach, sustainable development is built on resource harmony and valuation. This produces a system of interdependence between our surroundings and socioeconomic progress [43]. In this paper, a theoretical framework involving CBC and the local approach is applied to dynamic BRI strategies in AFTZ to explain the dynamic interaction between spatial structure, economic shifts, and local responses. We also examine what lessons can be derived from it to better address the challenges of CBC within inner SEZs. Focusing on regular SEZ cases provides an excellent opportunity to evaluate the regional challenges of CBC. These provide multiple perspectives on CBC and BRIs. Additionally, BRI projects provide insights into how economic and spatial restructuring affects the implementation of CBC. Based on the previous academic studies in the literature regarding FDI [44–47], public participation, impact assessment [48], spatial planning [49], and broad-range evaluations [50–53], we formulated a model to conduct a productive and integrated investigation of CBC [54].

The model was formulated through the following factors: driving (D) factor, pressure (P), states (S), impact (I), and response (R), i.e., the DPSIR framework. The internal relationships and interactions between components are shown by the DPSIR framework [52]. This framework was used because of its high potential to cover the diversity of social, economic, and spatial sustainability issues, including urbanism [52,55]; environmental sustainability [53,56–58]; sustainable urbanism [52,55,56]; and socioeconomic influences on biodiversity, ecosystem services, and human well-being [59]. Additionally, due to its ability to incorporate the dynamism of the BRI strategies, local perspectives, and the ability to describe interactions between each part, according to the DPSIR paradigm, a series of causal relationships exist between the driving forces, pressures, states, impacts, and responses, affecting social, economic, and spatial aspects (Figure 1). The DPSIR has several disadvantages. For example, issues cannot be captured unless a continuous study of the same indicators is conducted at regular intervals to grasp their dynamics; society's viewpoint will influence how it reacts. As well as effectiveness, the impact of a reaction may also differ depending on the context [60].

This study adopts the following research methods: a field investigation, in-depth interviews, analyses, and mapping. The aforementioned DPSIR framework informed our field investigation. Further details are presented in Table 1.

Our data analysis entailed quantitative evaluation by employing Shannon entropy to calculate the weights of the indicators used. In addition, it involved indexing formulae, which is dependent on research objectives, and creating and implementing an indexing system to evaluate the complexity of CBC development. These data were then verified using a Genetic Algorithm (GA) and mapped using Geographic Information Systems based on relevant regional values.

Maintaining a strategic vision in more diversified cooperative enterprises may prove difficult as each participating region seeks funds for localized initiatives that benefit them. The COVID-19 pandemic yielded concerns about regional concentration and a lack of robustness and adaptability in supply networks [11]. Concerning BRI strategies, economic development, and spatial restructuring optimization challenges, our goals can be categorized as follows: maximizing drivers, reducing pressures, maximizing impact, and minimizing responses. The applied DPSIR framework is presented in Figure 1.

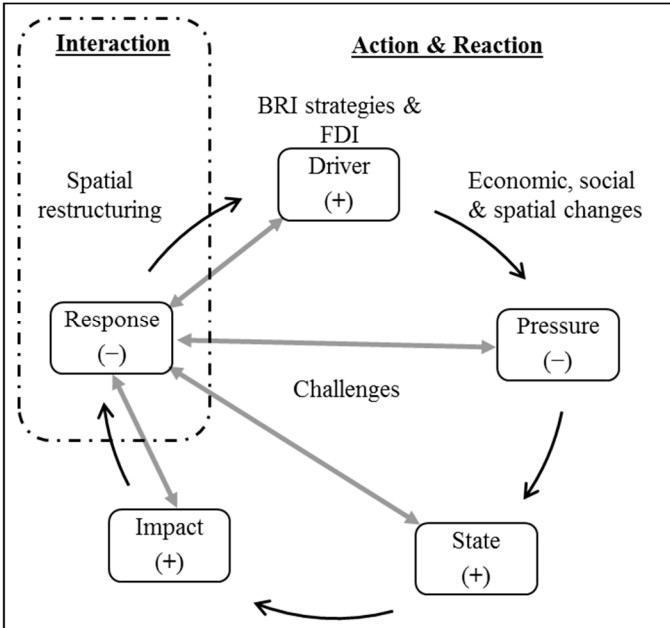

**Figure 1.** Applied DPSIR framework: AFTZ's transformation from the perspective of locals.

**Table 1.** DPSIR, number of interviewees, key points raised, and their purposes/insights.

| DPSIR | Number | Key Points and Their Purposes/Insights |
|---|---|---|
| Drivers | 7 | The driving power of BRI-driven FDI and economic changes on spatial development, BRI strategies, capital entry process, economic vitality and performance, technology transfers, etc. |
| Pressure | 15 | The pressure of economic changes on spatial development, livability, transit, land availability and pricing, spatial quality and development, housing, changes in the population, etc. |
| States | 4 | States of economic changes and spatial development, sort, combination, and geographic location of spatial restructuring based on local approaches; the perception of local on social, economic, and spatial changes, etc. |
| Impact | 9 | Impact of economic changes on spatial development and the impact of AFTZ transformation. Operating conditions, the composition of local groups, the relationship with the project-leading enterprise, the benefits and damages of AFTZ development, the use of land, changes in AFTZ space, etc. |
| Response | 6 | Responses to economic changes and spatial development; attitudes to spatial restructuring, incentives, spatial regulation, etc. |

For the present study, we considered social, economic, and spatial data while analyzing the first-hand data we gathered via our fieldwork and dissemination of questionnaires. The assessment model used also offers a tool for the comparative and comprehensive study of various regions in the five sections of the AFTZ. In this paper, the results of our study are compared thoroughly, and various methods are used to enhance each result. Additionally, from our results, decision-making and strategic advice can be provided for CBC practices in BRI. Further, our results can be used to assess potential CBC and countermeasures in the AFTZ and other regions after the COVID-19 pandemic [11].

## 4. Materials and Methods

Based on the aforementioned theoretical framework intended for use to assess CBC in AFTZ's spatial transformation and planning, this section presents the study area, dataset, DPSIR framework, indexes, and Genetic Algorithm (GA) optimization and mapping. Dynamic SEZ spatial restructuring was evaluated using a model that takes into account the effects of economic change on SEZ spatial restructuring. The DPSIR framework evaluates indicators including drivers, pressures, states, impacts, and responses for BRI-driven FDI. Our field investigation involved in-depth interviews about CBC transition in the study area. The GA was used for the verification of determined indexes.

### 4.1. Study Area: General Situation and Importance of the Study Area

This paper focuses on Iran's regular SEZ cases for three reasons: Iran is first and foremost one of China's most significant economic and strategic allies along the BRI. The AFTZ's role as SEZs and as part of the BRI strategy also allows for performance evaluations. The AFTZ is located in northwest Iran, on the northern border of the East Azerbaijan province, between 45°15′ and 47°28′ East longitude and 38°50′ and 39°17′ North latitude. Despite their significance, development in areas remote from economic activities near country borders has been overlooked [61]. The Aras River connects the province to the republics of Nakhichevan, Armenia, and Azerbaijan (Figure 2).

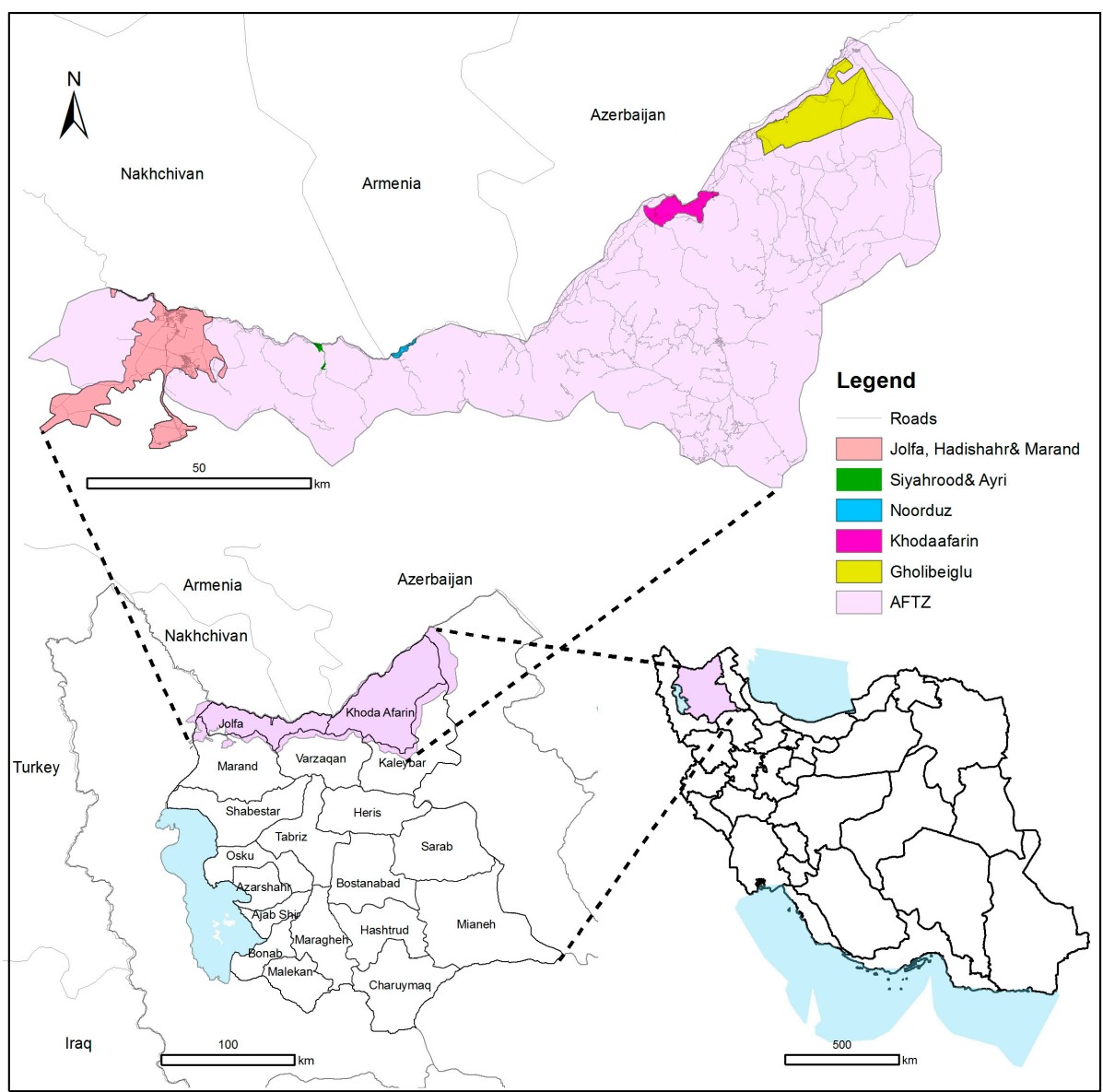

**Figure 2.** Map of the AFTZ (bordering the East Azerbaijan province).

According to the latest resolution of the Board of Ministers for the development of AFTZ, No. 79918/T 56509 AH, the region covers an area of approximately 71,848 hectares (ha). AFTZ consists of five sections: Jolfa, Hadishahr, and Marand 40,194 ha (S1), Siahrood-Ayri 332 ha (S2), Noorduz 567 ha (S3), Khodafarin 5680 ha (S4), and Golibeiglu 25,075 ha (S5). Agriculture, forestry, residential land, and industrial facilities are the primary land uses. The AFTZ has convenient access to Tabriz and Ardabil International Airports and intersects with the Tabriz–Aslandoz axis, which connects the AFTZ's entry and exit points with neighboring countries. Due to the AFTZ's proximity to the South Caucasus and the Commonwealth of Independent States (CIS), as well as the Jolfa Railway Station in the east, rail transport capacity is high. In general, it has the combined advantages of rail, aviation, and road transportation [62]. The AFTZ has attracted national and international attention due to its distinct strategic and economic position with neighboring nations and resourceful cities—Tabriz, Ardabil, and Marand—as well as its role as SEZs in BRI strategies.

### 4.2. In-Depth Interviews

In-depth interviews involve developing a plan, preparing materials (such as questions and maps of relevant sections), conducting the actual interviews, and evaluating the results. We conducted in-depth interviews with various community groups and residents about spatial restructuring and economic changes through asking interview questions (in Persian) informed by the drivers, pressures, states, impacts, and responses to BRI strategies after 2013 in the AFTZ to understand individuals' approaches to strategy application and gather responses from communities in the AFTZ to understand what they consider to be beneficial CBC interactions. A total of 322 local family heads (primarily men) voluntarily participated in in-depth interviews and were randomly selected. The average time taken to provide answers to our questions was 40 min. In this study, individuals considered important to the research purposes were interviewed in-depth, and the collected data were a primary source for demonstrating the spatial distribution patterns of regional approaches to spatial planning and territorial governance.

### 4.3. Fieldworks

We went to AFTZ for fieldwork from August 2020 to September 2021, mainly observing the physical landscape and spatial restructuring of AFTZ. The interviews focused on spatial development, land use, the FDI entry process, BRI strategies, AFTZ spatial change, the relationship between various subjects, and planning. In the process of our field investigation, we observed and recorded events in regional spaces to better understand the area overall. This helped us understand the local culture and customs. It helped improve our interview questions by allowing us to add more details and improved interview efficiency. Additionally, we collected planning and policy texts through the network and field channels to better understand the AFTZ's developmental context, spatial transformation, and reconstruction orientation. Considering the representativeness and comprehensiveness of the interviewees, a total of 326 households from all over the AFTZ participated, with S1 (137), S2 (25), S3 (27), S4 (43), and S5 (94), 95% confidence level, and a 2.65 percent margin of error. Descriptive statistics for individuals are presented in Table 2.

**Table 2.** Descriptive statistics for individuals (n = 322).

| Individual Level Variables | n (%) | | n (%) |
|---|---|---|---|
| Gender | | Number of co-occupants | |
| Male | 298 (92.55%) | Less than three | 28 (8.70%) |
| Female | 9 (2.80%) | Four or five | 142 (44.10%) |
| Missing | 15 (4.66%) | More than five | 10 (3.11%) |
| Age (years) | | Single | 64 (19.88%) |
| 20 to 30 | 50 (15.53%) | Missing | 78 (24.22%) |
| 31 to 40 | 107 (33.23%) | Monthly income level * | |
| 41 to 60 | 105 (32.61%) | IRR 0–1,000,000 (USD 0–77.41) | 1 (0.31%) |
| Missing | 60 (18.63%) | IRR 1,000,000–2,000,000 (USD 77.42–154.82) | 12 (3.73%) |
| Residing duration | | IRR 2,000,000–3,000,000 (USD 154.823–232.23) | 64 (19.88%) |
| 1 to 5 | 2 (0.62%) | IRR 3,000,000–4,000,000 (USD 232.24–309.65) | 7 (2.17%) |
| 5 to 10 | 1 (0.31%) | IRR 4,000,000+ (USD 309.65+) | 7 (2.17%) |
| More than 10 | 296 (91.93%) | Missing | 232 (72.05%) |
| Missing | 23 (7.14%) | The primary source of income | |
| Education | | Trade | 53 (16.46%) |
| Primary school | 1 (0.31%) | Industry | 81 (25.16%) |

**Table 2.** *Cont.*

| Individual Level Variables | n (%) | | n (%) |
|---|---|---|---|
| Middle school | 1 (0.31%) | Service | 119 (36.96%) |
| High school | 25 (7.76%) | Agriculture, Aquaculture, and livestock | 23 (7.14%) |
| Collage/university | 234 (72.67%) | Missing | 99 (30.75%) |
| Missing | 62 (19.25%) | | |
| Residency situation | | | |
| Own | 144 (44.72%) | | |
| Kinsfolks | 67 (20.81%) | | |
| Rented | 69 (21.43%) | | |
| Missing | 42 (13.34%) | | |

\* Source: based on data from the Central Bank of Iran and authors' estimation.

### 4.4. Measurement of Regional Variables: Index Calculation of AFTZ

Based on the DPSIR framework shown in Figure 1, which we applied to examine the acquired dataset (presented in Table 1) from local's in-depth interviews at the AFTZ, the ruling matrix needs to be normalized to eliminate the effect of different units and measures between attributes. Shannon entropy was used to determine each index gravity based on the index weight [51,63–66] and to eliminate inconsistencies in the questionnaire's unit dimensions and equivalence judgments [52,64,67]. Data normalization was performed on certain parts of the questionnaire's response matrix to obtain research outcomes as well. The index layer association for the Socio-Spatial Development Index (SSDI) model is established based on the following Equation (1):

$$SSDI = \frac{D_i \times S_i \times I_i}{P_i \times R_i} \tag{1}$$

the D, P, S, I, and R for the SSDI were calculated in the following form:

$$R_{D_i} = \frac{F_{D_i}}{n_D} \tag{2}$$

$$P_{D_i} = R_{D_i} \times 100 \tag{3}$$

In this equation, $R_{D_i}$ and $F_{D_i}$ are the relative frequency and frequency of factor i, respectively. At the driver indicators, $n_D$ is the number of observations. Additionally, for Equation (3); $P_{D_i}$ and $R_{D_i}$ are the percentage frequency and relative frequency, respectively—where i = 1, 2, 3 ... n.

$$D = \sum_{i=1}^{n} W_{D_i} \times P_{D_i} \times K_{D_i} \tag{4}$$

$W_{D_i}$ is the weight of $K_{D_i}$, $P_{D_i}$ is percentage frequency of factor i, $K_{D_i}$ is a numeric value for the driving force index i. Here, i = 1, 2, 3 ... n. This calculation was repeated for each of the pressure (P), states (S), impact (I), and respond (I) layers in Equation (4) to determine results.

**Data set adjustment: reconstructed index values and feasible uplift**

As illustrated in Figure 1, the applied framework for the development of and spatial restructuring in the AFTZ was necessary to the calculate Restructured Socio-Spatial Development Index (RSSDI) to determine the maximum feasible index values. Thus, to calculate the RSSDI, the maximum values for the driving force ($D_{max}$), states ($S_{max}$), and

impact ($I_{max}$), as well as the minimum values for pressure ($P_{min}$) and response ($R_{min}$), were determined and recalculated via Equation (5).

$$\text{RSSDI} = \frac{D_{max} \times S_{max} \times I_{max}}{P_{min} \times R_{min}} \tag{5}$$

$$\text{Feasible uplift} = \text{RSSDI} - \text{SSDI} \tag{6}$$

Then, we determined feasible uplift at AFTZ restructured index values subtracted from the current index values provided by Equation (6) to determine results.

### 4.5. Genetic Algorithm: Optimal Values for Most Suitable Spatial Development

The GA identified the fittest candidates in a complex space of solutions based on random information exchange. The fitness function of GA in MATLAB R2021b was used to determine optimal values and the validation process. The primary goal is to achieve a balance between efficiency and effectiveness in a variety of situations [68]. Using scatter plots were generated using the standardized and normalized linear response equations and GA fitness values presented in Tables 1 and 3. In the optimization algorithm used by the GA to calculate fitness, evaluate outcomes, and analyze dynamic interactions within the DPSIR framework, each individual's fitness is calculated using Equation (7). This corresponds to the simulation results of D, P, S, I, and R optimization (shown in Figure 1).

$$\text{fitness} = \max \sum_{i=1}^{n} (D_i \times S_i \times I_i)/(P_i \times R_i) \tag{7}$$

**Table 3.** Indicators and weights used in the DPSIR framework.

| Criterion | Indicator | Weights |
|---|---|---|
| Drivers | | |
| | Driving power of economic changes on spatial development | 0.0093 |
| | Export-driven business and industry growth | 0.0090 |
| | Equity and Income GDP | 0.0116 |
| | Economic vitality and development | 0.0176 |
| | Economic performance | 0.0247 |
| | Technology transfers for Industrial structure | 0.0230 |
| | FDI and Strengthening of Enterprises | 0.0280 |
| Pressures | | |
| | The pressure of economic changes on spatial development | 0.0322 |
| | Planning Communities for all age | 0.0130 |
| | Livability level and pleasant public Spaces | 0.0138 |
| | Transit and pedestrian accessibility | 0.0242 |
| | Green space area | 0.0278 |
| | Built form representation | 0.0321 |
| | Spatial quality in public spaces | 0.0314 |
| | Local spatial growth | 0.0170 |
| | Greening coverage in a built-up area | 0.0244 |
| | Spatial organization | 0.0263 |
| | Land consumption per capita | 0.0355 |

**Table 3.** *Cont.*

| Criterion | Indicator | Weights |
|---|---|---|
| Pressures | | |
| | New housing developments | 0.0177 |
| | Land cost | 0.0098 |
| | Availability and pricing of lands | 0.0340 |
| | Affordable housing | 0.0304 |
| States | | |
| | States of economic changes and spatial development | 0.0660 |
| | Sort of spatial development | 0.0176 |
| | Combination of spatial development | 0.0350 |
| | Location of spatial development | 0.0508 |
| Impacts | | |
| | The impact of economic changes on spatial development | 0.0165 |
| | Inequality level | 0.0100 |
| | Geographic Distribution of FDI | 0.0208 |
| | Locals' well-being | 0.0201 |
| | Population growth | 0.0247 |
| | Depopulation and emigration from the region | 0.0449 |
| | Social capital (networking, shared values, and trust) | 0.0346 |
| | Social Cohesion and regional identity | 0.0193 |
| | Public participation in the decision-making process | 0.0224 |
| Responses | | |
| | Response to economic changes and spatial development | 0.0143 |
| | Access to land resource | 0.0184 |
| | Environmental pollutions | 0.0180 |
| | Social Traditions, beliefs, and Identity | 0.0222 |
| | Emigrants and expatriates' community | 0.0238 |
| | Other locals' interest | 0.0277 |

*4.6. Spatial Visualization and Mapping*

The kriging tool in the Spatial Analyst tool in ArcMap 10.4® was employed for the ordinary kriging procedures. Kriging takes into account the spatial arrangement of the sample points because weights vary depending on how the samples are arranged. Kriging works best with evenly distributed data. Autocorrelation was assessed via kriging. Ordinary kriging is the most commonly used method for exploring field data [69].

*4.7. Data Analysis*

A total of 322 in-depth interviews were conducted. Each of the BRI-driven economic changes and regional performances in the AFTZ section were assessed using 41 inputs in the DPSIR framework. This yielded an overall frequency for local approaches, and spatial restructuring originated from BRI-driven FDI through CBC. Integrated DPSIR framework, Shannon entropy, index values, and a Genetic Algorithm were used to examine whether economic change increases or decreases the dynamism of regional balances. In addition, local reactions to spatial restructuring were either in favor of or opposed to planned CBC strategies for spatial development in AFTZ.

## 5. Results

This study included 322 households living in AFTZ and its surrounding areas. Integrated Shannon entropy was used to determine index weights based on in-depth interviews. Empirical findings, including summary statistics, estimated values for drivers, pressure, states, impact, responses, SSDI, GA optimization, and RSSDI, are depicted. AFTZ spatial restructuring and balanced development challenges were discussed, along with an analysis of different sections for maximum viable uplift and local approaches to spatial restructuring.

*Empirical Outcomes*

By modeling socio-environmental interdependencies, decision-makers can draw comprehensive conclusions that support a more sustainable future [70]. Using in-depth interviews with residents of the AFTZ, we determined the indicators of the local approach to economic growth and spatial restructuring for each of the drivers, pressures, states, impacts, and responses of CBC and BRI since 2013. Table 3 shows the total number of 41 selected indicators and their respective weights in the DPSIR framework.

Social, spatial, and economic indicators are all assessed using roughly the same number of indicators, but their weights are almost the same; the weights of economic indicators are lower in the CBC assessment. A statistical summary of the study method, including applied linear equations, mean, standard deviation, minimum, maximum, and weights, determined optimal values via GA and DPSIR tendencies (presented in Table 4). High differences between the drivers, states, and response groups, as well as between the pressure and impact groups, created a challenging dynamic between positive and negative tendencies in the DPSIR framework. As a result of negotiation and bargaining, common rules, both formal and informal, have developed for mutual benefit in addition to hegemony [71]. This is necessary to yield balanced CBC and facilitate the development of the AFTZ.

**Table 4.** Summary statistics.

|          | Linear Equations | Mean | SD | Min | Max | Weights | Optimal | Tendencies |
|----------|------------------|------|------|--------|--------|---------|---------|------------|
| Drivers  | D = 0.2499 × P + 0.0084 | 0.012 | 0.007 | 0.0009 | 0.0443 | 0.178 | 0.019 | + |
| Pressure | P= 0.037 × S + 0.012 | 0.011 | 0.008 | 0.0005 | 0.0454 | 0.371 | 0.012 | − |
| States   | S = 0.7377 × I + 0.023 | 0.013 | 0.006 | 0.0013 | 0.0288 | 0.082 | 0.044 | + |
| Impact   | I = 0.1541 × R + 0.0065 | 0.033 | 0.014 | 0.0007 | 0.0682 | 0.253 | 0.017 | + |
| Response | R= 0.125 × D + 0.0099 | 0.012 | 0.006 | 0.0006 | 0.0269 | 0.147 | 0.010 | − |

Note: The observations (n = 322) cover a period from 2013 to the present day.

However, due to the ongoing evolution of economic drivers in the CBC transition process, as determined by calculated SSDI, the overall impact of economic changes on AFTZ spatial restructuring has come under increasing pressure (check Table 4).

Large differences in drivers, states, and response groups, as well as pressure, impact, and drivers, impose dynamic challenges between positive and negative tendencies. In terms of the identified drivers, states, and impact values, S1 and S5 appear to be highly and moderately privileged, whereas S2, S3, and S4 relate to underprivileged regions, which indicates a rise in the prominence of regional imbalances. Table 4 reports regional empirical results for CBC via DPSIR on economic changes and the spatial restructuring of the AFTZ. It displays estimation results in columns (1) through (7). Despite dynamic interactions among drivers, pressure, and state values, impact values have a strong presence. As a result of the low values associated with responses to economic changes and spatial restructuring, planned BRI strategies and CBC seem more popular with locals. Table 5 presents the index results for the SSDI, RSSDI, and DPSIR values and the percentage of maximum feasible gain for AFTZ and its specific sections based on local responses to the spatial restructuring.

**Table 5.** Determined sums of DPSIR, SSDI, and RSSDI values in the AFTZ.

|  | D | P | S | I | R | Index | Sum |
|---|---|---|---|---|---|---|---|
| | 1.74 | 1.84 | 2.21 | 5.29 | 1.86 | SSDI | 18.10 |
| S1 | 2.59 | 1.6 | 2.58 | 2.26 | 1.33 | Optimal | 16.27 |
| | 6.75 | 4.46 | 3.99 | 7.42 | 0.71 | RSSDI | 67.56 |
| | 0.31 | 0.20 | 0.38 | 0.64 | 0.38 | SSDI | 1.03 |
| S2 | 0.49 | 0.30 | 1.11 | 0.43 | 0.25 | Optimal | 3.06 |
| | 1.15 | 0.57 | 0.79 | 0.94 | 0.14 | RSSDI | 10.47 |
| | 0.34 | 0.34 | 0.44 | 1.02 | 0.33 | SSDI | 1.36 |
| S3 | 0.51 | 0.31 | 1.15 | 0.44 | 0.26 | Optimal | 3.18 |
| | 0.97 | 0.79 | 0.88 | 1.40 | 0.13 | RSSDI | 11.27 |
| | 0.48 | 0.34 | 0.45 | 1.19 | 0.41 | SSDI | 3.14 |
| S4 | 0.84 | 0.52 | 1.90 | 0.73 | 0.43 | Optimal | 5.26 |
| | 1.48 | 0.93 | 1.03 | 1.65 | 0.18 | RSSDI | 19.43 |
| | 1.10 | 1.03 | 0.58 | 2.33 | 0.76 | SSDI | 4.08 |
| S5 | 1.83 | 1.13 | 4.16 | 1.60 | 0.94 | Optimal | 11.50 |
| | 3.32 | 2.25 | 1.90 | 3.68 | 0.40 | RSSDI | 33.49 |

Bottom-up initiatives will be considered ineffective under a top-down strategy if they do not consider or incorporate the capabilities and needs of the local community [32].

In general, the spatial distribution of D, P, S, I, and R shows a high dependence on drivers' geographical locations; however, this phenomenon increases concentration, which causes distortions in inner-SEZs dynamics. A spatial demonstration of calculated D, P, S, I, and R values for separate AFTZ sections is depicted in Figure 3. Indicators of economic progress and their spatial influence on regional spatial structures [36,37], as confirmed by evidence from DPSIR maps, demonstrate that CBC is largely effective in spatial restructuring; however, the increased concentration of BRI-driven FDI causes dynamic interactions within SEZs to be imbalanced.

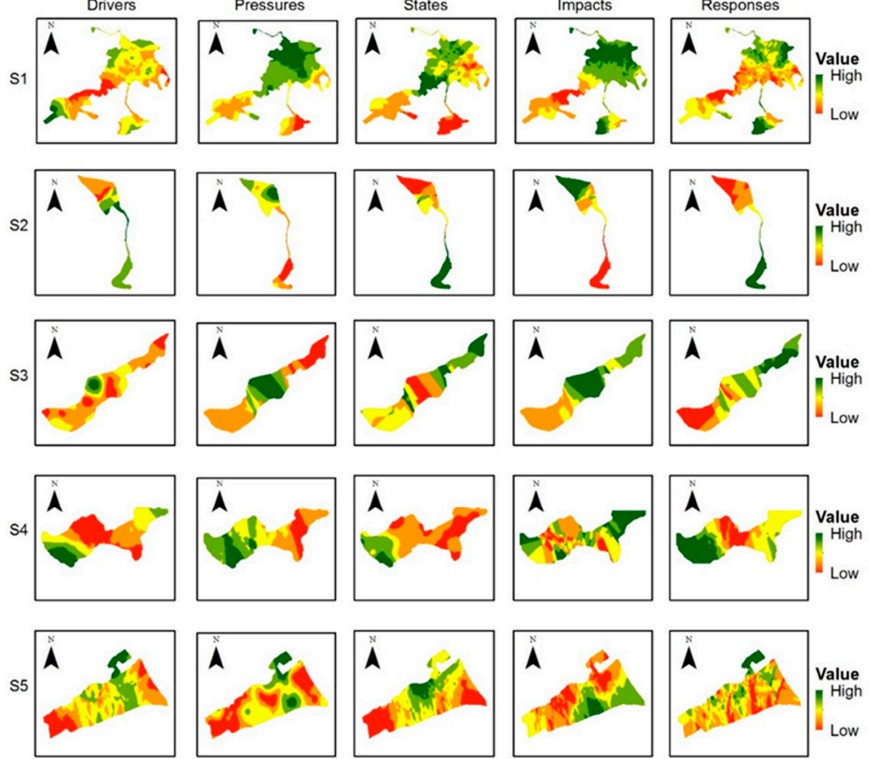

**Figure 3.** Spatial demonstration of calculated D, P, S, I, and R values for separate areas in the AFTZ.

As demonstrated in Figure 4, determined values for SSDI spatial concentration and divers confirm inner SEZs and separate areas' dependency on CBC as a main cause for imbalances in economic and spatial restructuring.

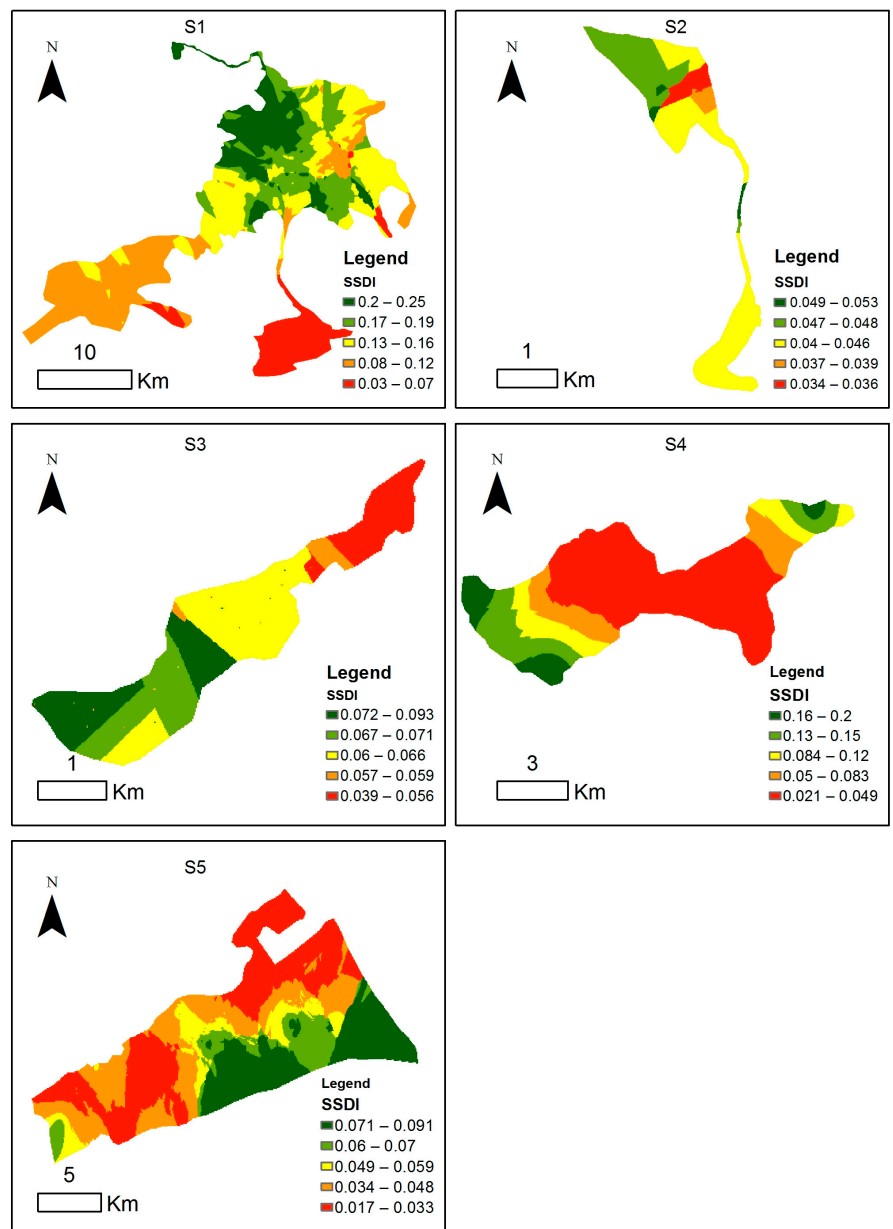

**Figure 4.** Spatial demonstration SSDI values calculated for areas in the AFTZ.

The results regarding RSSDI revealed a maximum potential 5.13-fold increase in SSDI values in the entire AFTZ (see Table 4). RSSDI is used to compare SSDI and make maximum feasible change estimates to see how it can significantly improve D, S, and I estimation efficiency. In addition, it reduces the effect of P and R on CBC-related issues caused by FDI resulting from BRI. The maximum possible changes in DPSIR values for each of the five AFTZ sections after subtracting RSSDI from SSDI are shown in Figure 5. As shown in S1, S3, and S4, the P value is higher or equal to the state's values. On the other hand, the P values in S2 and S5 are slightly lower than the S values, indicating that the AFTZ sections and inner SEZs are imbalanced or on the verge of being imbalanced.

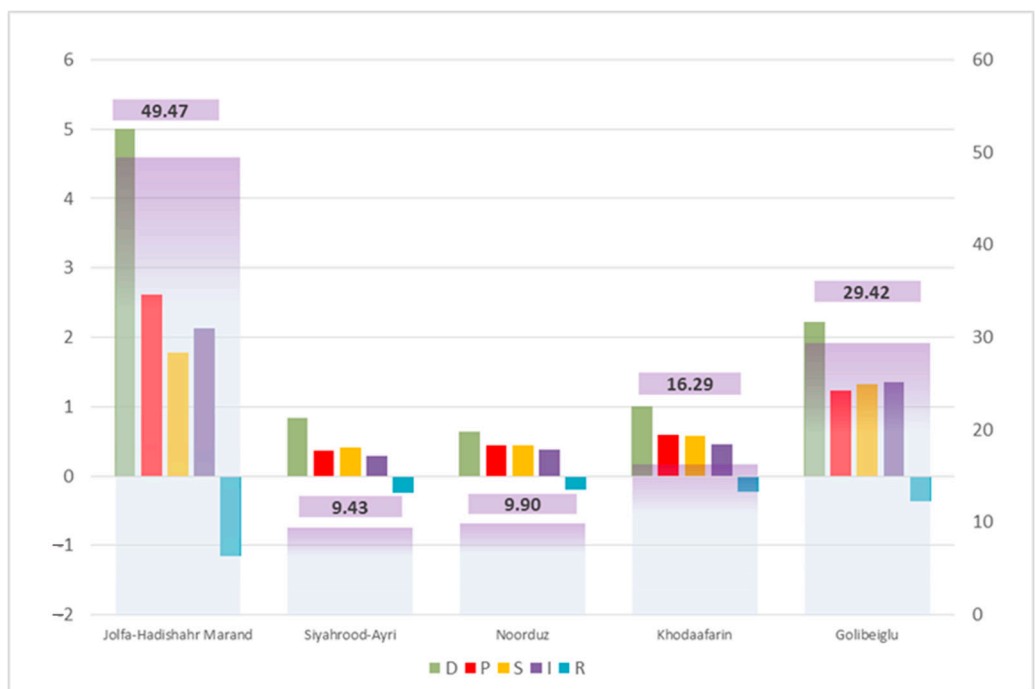

**Figure 5.** The maximum feasible changes for DPSIR values for each of the five AFTZ areas. The feasible change values are represented by the X-axes value on the right side.

The results in AFTZ showed spatial heterogeneity, which, paradoxically, has tended to increase spatial inequalities [32], land use conflict [33,34], and inconsistency [72,73]; hence, careful planning is required. Any intervention that focuses solely on physical upgrading will have little influence; instead, a place syntax is required to communicate notions such as social interaction, networks, or social ties, much as it was for settlement transformation [74].

To achieve a balance between efficiency (high D, S, and I) and effectiveness (low P and R) in a variety of situations (DPSIR), based on Equation (7), our in-depth interview findings, the GA, we identified optimal values for D, P, S, I, and R (depicted in Table 3) and determined sums for each section of the DPSIR framework (presented in Table 4). Comparing DPSIR values with determined thresholds, S1, S2, and S3 showed imbalanced results, with 55%, 53%, and 56% passing the optimal threshold, respectively. In contrast, S4 and S5, with results that were 62%, and 57% under the optimal threshold, respectively, showed balanced CBC and development. For the areas in the AFTZ, the determined outcomes were over the calculated optimal values presented in Figure 6. A spatial performance disparity analysis of BRI-driven economic changes since 2013 demonstrates that regional imbalances increase (S1, S2, and S3) as economic changes induced by CBC increase, as well as whether local responses to spatial restructuring promote common planning and strategic planning (S4, S5). This illustrates how local approaches and BRI strategies interact dynamically. Our study of disparities in the regional spatial performance of BRI strategies in AFTZ since 2013, which was supplemented by conducting in-depth interviews with locals, highlights the importance of spatial restructuring and economic changes in sustainable regional growth issues.

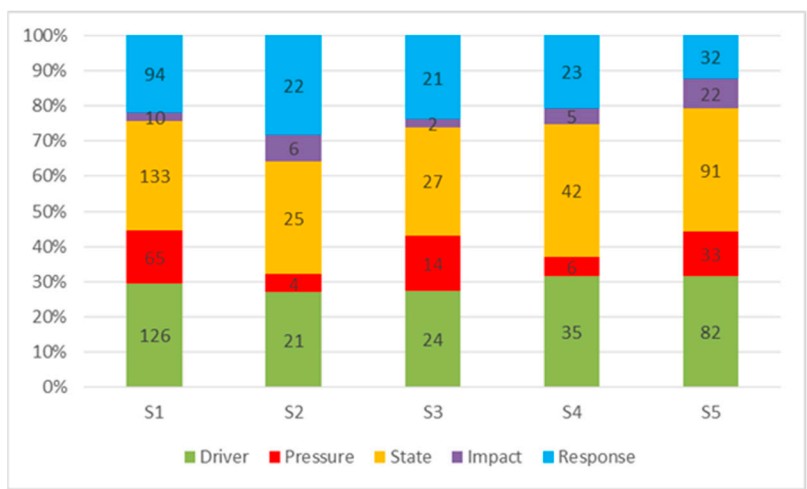

**Figure 6.** Determined outcomes for the AFTZ areas that were over the calculated optimal values.

## 6. Discussion

The CBC and transition process influenced the spatial restructuring of the AFTZ and the local approach to BRI-driven FDI. Regional imbalances are increasing due to economic development and spatial restructuring. Locals continue to prefer BRI strategies despite mounting pressure on AFTZs. According to the relevant research results, in terms of the influence of BRI strategies on the local approach to the spatial restructuring of the AFTZ and its encompassing areas, the five noted categories of inputs were ordered as drivers, pressures, states, impacts, and responses, and we established the final comparative DPSIR framework.

By modeling socio-environmental interdependencies, decision-makers can draw comprehensive conclusions that support a more sustainable future [70]. Through conducting in-depth interviews with residents of the AFTZ, we derived complete indicators for the local approach to economic growth and spatial restructuring for each of the drivers, pressures, states, impacts, and responses of CBC and BRI since 2013.

High differences between the drivers, states, and response groups, as well as between the pressure and impact groups, created a challenging dynamic between positive and negative tendencies in the DPSIR framework. As a result of negotiation and bargaining, common rules, both formal and informal, have been developed for mutual benefit in addition to hegemony [71].

Rapid economic transformation and socio-spatial uplift are multifaceted challenges. Due to competitive local interests, economic changes, and spatial restructuring, policies may harm the economy, industry, and sustainable development. The relevant policies need restructuring to achieve the targeted outcomes [24], and regional growth [28] and spatial planning could act as a conduit for this. There is a dynamic relationship between economic changes and their spatial manifestation [36], highlighting the significance of social and perceptual data [3] while carrying out spatial restructuring plans, as well as providing crucial documentation for future international cooperation by investigating and harmonizing local engagement and sustainable development [74].

The importance of the local context for the business sector cannot be underestimated. The proposed procedures that incorporate collaboration with local communities are expected to create new development opportunities for investors to attract international investment [75]. As well as regional spatial development [76,77], the geographical distribution of public resources based on the need [75] to reduce regional disparities [32], project uncertainties, and unwanted societal side effects [19] is also required.

The developed framework demonstrated the feasibility of incorporating local approaches to spatial restructuring and planning in areas in the AFTZ. As well as fulfilling other requirements, such as collaboration and coordination, this results in the effective match-

ing of service expectations, spatial equity, and improved public performance [75,77–79], preventing radical changes [19], regional disparities [32], and enhancing economic ties [29]. The spatial aspects of economic changes [36] had significant influences on spatial restructuring in AFTZ and heterogeneity, with land use types being one of the primary drivers, which could be characterized as land use conflict [33,34].

Population and the presence of private capital appear to promote effective local economic restructuring [19]. Spatial restructuring based on bottom-up interactions resulted in an equivalent change and reduced regional disparities [32] in the AFTZ sections, indicating that FDI spurred by the BRI has a favorable impact on CBC development (see Table 4). Hence, top-down policy incentives in AFTZ like FDI, along with bottom-up collaboration and execution, according to the applicable approach, would contribute to the success of the BRI strategy (check Figure 5) and enable higher levels of governance and CBC to grow stronger [77].

When the RSSDI was superimposed, the pressure and responses of each section decreased, while the impact and states increased along with an increase in the influence of the drivers, as defined in the methodology. Regional economic policy is significantly impacted by regional disparities and their short- and long-term changes [19]. Companies expanding abroad, as well as investors in general, face the challenge of identifying valuable locations and sites to launch their operations [76], which necessitates observing locals and the demand for institutionalization in the context of local [73,74] and spatial analysis. It is worth noting that, in addition to the factors mentioned above, previous developments and construction, laws and regulations, and approvals [73] could also influence the study results.

Despite the existence of perspectives that are overlooked by obsessive administrators and community planners [78], an integrated stakeholder engagement method is necessary for development and implementation to unify information disclosure and community consultation activities [14], which this study partially covered. Feasibility analysis and resource mobilization are necessary as prerequisites for decision-making [72] and providing possible directions for development plans. Local governments, the general public, and non-governmental groups all contribute to and coordinate favorable public policies [30] for CBC in support of BRI strategies.

## 7. Conclusions

Focusing on regular SEZ cases provides several excellent opportunities for CBC to evaluate the challenges it faces on a regional scale. CBC, spatial planning, and territorial governance require regional impact assessments to deepen ties. Diverse approaches are centered on the dynamics of inner SEZs. These provide multiple perspectives on CBC to BRI. Additionally, BRI projects provide insights into how economic and spatial restructuring affect CBC implementation issues. In this paper, we investigated the relationships between economic changes and spatial restructuring with panel data from five AFTZ sections in Iran since 2013. The paper explores CBC determinants for spatial planning and territorial governance.

In particular, by more closely examining the effect of BRI-driven FDI via the DPSIR framework, the results reveal that regional imbalances related to spatial performance disparities are slightly over the optimal values, and due to the increased impact of CBC local responses to spatial restructuring, they promote common planning and strategies. These results provide proof for SEZs in Aras to continue their CBC and optimize economic changes and spatial restructuring. This will increase BRI-related FDI inflows. Further, the concentration of BRI-driven FDI leads to a dramatic imbalance of dynamic interactions within inner SEZs, which in turn leads to regional imbalances in AFTZs.

As CBC increases, regional imbalances increase as a result of economic development and spatial restructuring. Local resistance to strategic and common planning adds to the multifaceted CBC challenges and complications. CBC has strong potential to establish "coopetition," which is the act of cooperation despite competition, to find practical solutions

that benefit all parties. Multi-level coopetition should be emphasized for its importance in spatial and regional growth issues. Multi-aspect and statistical analyses can help to improve provision accuracy, close the gap between top-down and bottom-up approaches, raise awareness of socio-spatial syntax, and diversify related activities.

This research has the following implications for CBC investors and policymakers:

(1) Regardless of further specialized approaches to regional planning, to ensure alignment with CBC and planning strategies, it is imperative to develop comprehensive engagement among locals and include their interests. The economic and spatial performance of SEZs is strongly influenced by CBC, and this study provides a useful orientation for policymakers to understand that all CBC dimensions relate to local approaches, regional development, FDI-driven restructuring expansion in related actions, and economic development as a foundation for this regional growth. As a result, our research could have a significant impact on BRI projects and ensure the integrity of planning processes, helping to avoid radical effects.

(2) We approached the link between CBC and economic-spatial changes in SEZ from a local perspective. Here, financial and spatial restructuring is calculated based on the D-score, P-score, S-score, I-score, and R-score. The index system was implemented using in-depth interview data for 322 Iranian households, representing 5 distinct sections of CBC influence since 2013. In our analysis of social, economic, and spatial restructuring indicators, we found that improved levels of CBC and cooperation with locals are rewarded by higher acceptance of BRI strategies. Through collaboration, this research contributes to the CBC debate by showing that competition and cooperation complement one another. However, we found that there were fewer challenges and levels of resistance associated with FDI.

(3) Preliminary results indicate that regional imbalances have a negative effect on CBC. In particular, by more closely examining the effect of BRI-driven FDI via the DPSIR framework, the results reveal that regional imbalances due to spatial performance disparities are slightly over the optimal values, and due to the increased pressure of CBC and local responses (P-score and R-score) to spatial restructuring, they promote common planning and strategies for higher drivers, states and impact areas. These results provide evidence that SEZs should continue their CBC and optimize economic change and spatial restructuring. This will result in an increase in foreign direct investment. In case BRI-driven FDI leads to uncontrolled development, it creates dramatic imbalances within inner special economic zones that can negatively impact regional balances. For this reason, BRI strategies should prioritize the social and spatial dimensions of CBC over economic and industrial dimensions that directly affect stakeholders.

Regarding the limitations of this study, we only collected data from one SEZ. The use of multicenter datasets likely would have yielded better results. As a result of societal factors and limited participation, we evaluated only 322 interviewees, which is considered a fairly small sample size. However, in the future, we can improve the sample size and include more female participants. It should also be noted that the proposed model can be revised to focus on standard infrastructure projects, whereas future studies may target industrialized economies, where the developed model can be adapted.

Locals' areas of interest and weights were revealed during the in-depth interviews, and studies on other SEZs could produce different findings and be taken into consideration in the future to further study CBC, as well as factors such as size, neighbors, fragmentation, facilities, morphology, and patterns affecting spatial imbalances. This study covers the period before the COVID-19 pandemic; however, an updated study could reveal the flow of CBC during and after the pandemic, which would provide insight into the critical factors influencing the tolerance capacity of regions, although results may differ vastly following the COVID-19 pandemic.

**Author Contributions:** Conceptualization, H.J. and Y.F.; methodology, H.J. and Y.F.; software, H.J.; validation, H.J. and Y.F.; formal analysis, H.J. and Y.F.; investigation, H.J. and Y.F.; resources, H.J.; data curation, H.J.; writing—original draft, H.J.; writing—review & editing, H.J. and Y.F.; visualization, H.J.; supervision, Y.F. All authors have read and agreed to the published version of the manuscript.

**Funding:** This research received no external funding.

**Institutional Review Board Statement:** Not applicable.

**Informed Consent Statement:** Not applicable.

**Data Availability Statement:** Not applicable.

**Acknowledgments:** The authors would like to thank the editorial boards and blind reviewers for their help and guidance in the process.

**Conflicts of Interest:** The authors declare no conflict of interest.

## Nomenclature

| Symbol | Nomenclature |
| --- | --- |
| n | Number of interviewees |
| SSDI | Socio-Spatial Development Index |
| $D_i$ | Driver indicators |
| $P_i$ | Pressure indicators |
| $S_i$ | States indicators |
| $I_i$ | Impact indicators |
| $R_i$ | Response indicators |
| $D_{max}$ | Maximum feasible driving index |
| $S_{max}$ | Maximum feasible states index |
| $R_{max}$ | The maximum feasible response index |
| $R_{D_i}$ | Relative frequency |
| $F_{D_i}$ | Frequency of factor |
| $n_D$ | Number of observations |
| $P_{D_i}$ | Percentage frequency |
| $W_{D_i}$ | Weight of driving factor |
| $K_{D_i}$ | Driving force index numerical value |
| RSSDI | Restructured Socio-Spatial Development Index |
| $P_{max}$ | The maximum feasible pressure index |
| $I_{max}$ | Maximum feasible impact index |

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
