# Peer review of "Economic and Spatial Restructuring in the Aras Economic Zone: The Impact of Cross-Border Cooperation"

_sustainability, doi:10.3390/su151310289_

Round 1

Reviewer 1 Report

The topic seems interesting, yet the following comments are to be addressed. 

1. Title: The title is descriptive and conveys the main focus areas of the paper. However, it could be simplified to improve clarity and readability. Something like "Economic and Spatial Restructuring in the Aras Economic Zone: The Impact of Cross-Border Cooperation" might make it easier to understand at a glance.

2. Abstract: the abstract needs to be revisited to provide a concise summary of the research, including the objectives, methods, main findings, and implications. It should be accessible to a broad audience, not just specialists.

3. Introduction: the introduction needs to clearly outline the problem or issue, provide a review of relevant literature, state the research question or hypothesis, and explain the significance of the study. Please update the introduction section specifically lines 94-108. Also, the literature review must be comprehensive and up-to-date. It should also be critical, evaluating the strengths and weaknesses of previous work. Thus, consider adding the following references:

https://doi.org/10.1080/0042098966664

https://doi.org/10.1061/(ASCE)ME.1943-5479.0001003

https://doi.org/10.1080/17509653.2021.1991851

https://doi.org/10.1177/0275074020941699

https://doi.org/10.1016/j.gloenvcha.2021.102307

5. Theoretical Framework: the framework must be clearly explained. This includes the research design, data collection, data analysis, and any statistical methods used. It should also discuss any limitations of the chosen methods. Please illustrate more for lines 150-170.

6. Results and Discussion: the results must be clearly and logically presented. Consider using visuals such as graphs and tables can be very helpful here and update figures 3, 4, and 5. Also, the discussion should interpret the results in light of the research question, considering the implications and limitations of the findings. It should also suggest areas for future research. Please elaborate more on lines 390-404.

  1. 7. Conclusion: the conclusion must effectively summarize the key findings, state their implications, and suggest directions for future research. Please consider visiting lines 406-440. 

General Comments:

Since this paper is about a specific region (Aras Special Economic Zone in Iran), does the paper adequately provide context about this region? Information about the history, culture, and socioeconomic factors of the region could be useful for readers unfamiliar with it. Also, the paper should adequately explain how cross-border cooperation has influenced economic changes and spatial restructuring. The mechanisms of this influence should be clearly explained.

In the abstract, some sentences seem to be a comment on the manuscript rather than a part of the abstract. Consider revising these sentences.

Throughout the manuscript, ensure consistency in the use of tenses. For example, when discussing the methods, use the past tense to describe the processes that have been completed. 

Be cautious of the use of passive voice, as it can sometimes make sentences less clear. Where possible, use active voice for better readability.

When presenting the results, ensure that all tables and figures are properly labeled, and the text refers to them accurately.

Proofread the entire manuscript to correct the major grammatical errors or inconsistencies in punctuation.

Reviewer 2 Report

See the attachment.

See the attachment.

Reviewer 3 Report

Thank you for the opportunity of reading and reviewing your interesting manuscript. The paper addresses an important topic, although not much investigated, and the area under investigation is as well an area of much importance. The findings can be extended for other SEZ and for the case of other CBC. The manuscript is well structured and well written, with several revisions I consider it can really make a contribution in the area investigated.

I have several suggestions:

1. In the Introduction you should clearly indicate the research gap identified and the aims of your study. However, the contributions you made are more adequate in the final section (see line 97-102).

2. The Theoretical framework section should definitely be enhanced. You should have a deeper analysis of the theory in the field of regional development and CBC, including other zones like Europe, US, Russia, China etc. Here are several reading suggestions: Chirodea, F., Toca, C. V., & Soproni, L. (2021). Regional development at the borders of the European Union: introductory studies. Crisia, LI(Suppl. 1), 7-18. https://nbn-resolving.org/urn:nbn:de:0168-ssoar-76832-1 ; Ji, M.; Li, F.; Xu, S.; Zhuang, Y.; Bair, T.; Bilgaev, A.; Guo, K. Potential for Economic Transition and Key Directions of Cross-Border Cooperation between Primorsky Krai (Russia) and Jilin (China). Sustainability 202315, 4163. https://doi.org/10.3390/su15054163. Moreover, you should avoid referencing so many articles in one row, e.g. rows 124-125

3. Regarding the research itself, I find it as correctly conducted and well presented both in the sections regarding Method and in the section related to Discussion

4. In the final section I suggest to include not only the synthesis of the research but also to highlight the  theoretical and practical contributions, policy implications, and limitations of your research.

Good luck!

The language is fine, no issues detected.

Round 2

Reviewer 1 Report

All the reviewer's comments were addressed. The manuscript is ready for publication.